# Improving Broad Spectrum Blast Resistance by Introduction of the Pita2 Gene: Encoding the NB-ARC Domain of Blast-Resistant Proteins into Upland Rice Breeding Programs

Reny Herawati [1,*], Siti Herlinda [2], Dwi Wahyuni Ganefianti [1], Hendri Bustamam [3] and Sipriyadi [4]

1   Department of Crop Production, Faculty of Agriculture, University of Bengkulu, Jl. WR. Supratman, Kandang Limun, Kota Bengkulu 38122, Indonesia
2   Plant Protection Department, Faculty of Agriculture, Universitas Sriwijaya, Indralaya 30662, Indonesia
3   Plant Protection Department, Faculty of Agriculture, University of Bengkulu, Jl. WR. Supratman, Kandang Limun, Kota Bengkulu 38122, Indonesia
4   Biology Department, Faculty of Mathematics and Natural Sciences, University of Bengkulu, Jl. WR. Supratman, Kandang Limun, Kota Bengkulu 38122, Indonesia
*   Correspondence: reny.herawati@unib.ac.id

**Abstract:** Blast disease is generally more important in upland rather than lowland rice cultivation, representing one of the biotic obstacles in the development of upland rice. The objective of this study was to detect broad-spectrum blast resistance gene Pita2 encoding the NB-ARC (nucleotide-binding adaptor common in APAF-1, R proteins, and CED-4) domain of blast-resistant proteins in new upland rice lines from the breeding program for landrace rice varieties, with the goal of providing a novel source of blast-resistant germplasm for application in future upland rice breeding programs. In this study, we screened 19 inbred lines of landrace rice varieties challenged using local virulent isolates in greenhouse conditions and performed field evaluations to confirm blast resistance. Molecular analysis was conducted using six specific primers to detect broad-spectrum blast resistance, and sequence analysis was performed to detect the NB-ARC domain of blast-resistant proteins in the lines. Consistent results were observed between greenhouse screening and field evaluations, although there was variance in the level of resistance. The PCR assay showed that there were eight positive lines (G7, G8, G9, G11, G13, G14, G15, and G18) containing the Pita2 gene. Conserved domain analysis revealed that eight blast-resistant rice lines encode NB-ARC at sequence lengths ranging between 300 and 870 (450 bp). Using these sequences in BLASTX searching revealed 15 gene homologs of the eight rice lines, which were detected as Pita2 genes, with a similarity level of 81–99%. Further comprehensive studies should be performed to confirm the performance and resistance of candidate lines in field trials in various blast-endemic areas before being released as new upland rice varieties able to overcome the problem of blast disease in the field. In addition, the lines can also be used as a novel genetic resource in the blast-resistant upland rice breeding program on various rice cultivars.

**Keywords:** blast resistance; landraces; NB-ARC domain-containing protein; broad-spectrum resistance; Pita2 gene; upland rice lines

## 1. Introduction

The most significant upland rice disease is blast, which is caused by the fungus *Pyricularia oryzae* (Cooke) Sacc. In the tropics, blast disease generally has greater impacts on upland rice cultivation than lowland rice cultivation, representing one of the biotic obstacles in the development of upland rice. The rate of yield loss caused by blast disease in endemic areas reaches 50–100% [1]. In the Ciherang variety, the decrease in yield was 3.65 tons/ha or equivalent to 61% of the average production [2].

The chemical control of blast disease increases production costs. Therefore, the use of superior blast-resistant varieties is cheaper and more effective. The development of superior

varieties requires a different source of new blast-resistant genes. Thus, the availability of genetic diversity maintained in germplasm sources is an important factor in supporting rice breeding programs. The genetic background of superior varieties can be broadened through recommendations for landrace varieties in the breeding program [3,4].

Blast control through the assembly of rice varieties that have durable and polygenic resistance seeks to overcome blast pathogens that have multiple races and are very dynamic [5,6]. Therefore, the breeding program to obtain superior rice varieties is implemented through several stages: namely, determining the parents as the source of resistance genes, crossing between two or more elders, and evaluating the progenies. We have been conducting a breeding program since 2010, with the establishment of a baseline population and recurrent selection using Sriwijaya and Bugis as landrace varieties and IR 7858-1 and IR148 lines as donor parents of drought- and blast-resistant genes [7–9].

The evaluation of blast resistance in rice can be conducted in the field in endemic areas using a natural inoculum or in a screen house using an artificial inoculum [10]. Molecular detection of resistance genes can also be performed in these areas [3,11]. Furthermore, field evaluation can be conducted at a further stage to obtain varieties in the field resistant to various racial compositions because the varieties released to farmers will face multiracial environmental conditions [3]. Molecular detection is often conducted and helps breeders in selecting genotypes that have blast-resistant genes before being released as new varieties.

It is necessary to learn more about the molecular mechanisms involved in host–pathogen interactions and the strategic interrogation of resistance genes in cultivars, especially in terms of the identification of host resistance genes and pathogen avirulence genes. Currently, molecular markers are widely used to characterize collections of unrevealed gene banks, especially in untapped different allele resources [12,13]. However, the identification of broad-spectrum blast-resistant genes is critical for protecting plants against dynamic races of *P. oryzae*.

The genetically related virulence traits of *P. oryzae* have been studied by several researchers. There are several genes that have been cloned, namely *Pi37* on chromosome 1 [14], *Pib* on chromosome 2, *Pi9* [15] and *Pid2* [16] on chromosome 6, and *Pita* on chromosome 12 [17]. *Pup1* also contains genes for dirigent-like, fatty acid α-dioxygenase and aspartic proteinase, which govern the production of proteins that are critical in lignin biosynthesis, thereby affecting the hardness of plant cell walls [18]. Ref. [19] used 11 major blast resistance genes of rice (*Pi-d2, Pi-z, Piz-t, Pi-9, Pi-36, Pi-37, Pi5, Pi-b, Pik-p, Pik-h*, and *Pita2*) to identify 32 accessions resistant to *P. oryzae* using molecular markers. Previous studies revealed that *Pita2* have provided a broad spectrum for blast resistance compared to the *Pita*, and it is located at the centromere region of chromosome 12 [20,21]. Some of the cloned R genes encode proteins containing a nucleotide-binding site-rich repeat domain (NBS-LRR) [5,22]. Recent studies revealed that *Pita2* encodes a novel R protein unique to *Pita*, which is exactly the same as the previously cloned *Ptr* [23]; furthermore, it was found that *Pita2* rather than *Pita* was responsible for the specificity for some of the differential isolates with *AvrPita*.

Resistance proteins (R) in plants are involved in pathogen recognition and the subsequent activation of the innate immune response. Most resistance proteins contain a central nucleotide-binding domain. This is known as the NB-ARC domain and consists of three subdomains: NB, ARC1, and ARC2 [24,25]. The NB-ARC domain is a functional ATPase domain, and its nucleotide-binding state is proposed to regulate R protein activity. The nucleotide-binding site-leucine-rich repeat (NBS-LRR) resistance gene is the most decisive in determining the plant defense response, thus comprising the most predominant family of plant resistance genes [26,27]. Gene expression occurs based on the recognition of proteins encoding for components of the immune system in plants. The core nucleotides in the NB-LRR protein are part of the NB-ARC domain due to their presence in APAF-1 (apoptotic protease-activating factor-1), protein R, and CED-4 (*Caenorhabditis elegans* death-4 protein) [28]. The presence of effectors encoded by pathogens promotes intramolecular interactions involving NBS-NLR, among others, which triggers the oligomerization of

NLRs due to the substitution of ADP by (d) ATP in the NB-ARC domain. The induction of oligomerization is a well-known phenomenon triggered in animal NLRs [29,30]. NB-ARC has been revealed as the main domain facilitating NLR self-association in animal NLR structural analysis [31]. By contrast, there is a lack of knowledge about the decisive domain involved in the oligomerization of plant NLRs.

There is limited research on the structure of blast-resistant-gene-encoded proteins. Among them is the characterization of proteins from *Pita* and *Pi54*, including the downstream interaction partners of plant NBS-LRR proteins [32]. It is known that the NBS-LRR protein is involved in plant defense mechanisms.

The aim of this study was to detect the presence of the broad-spectrum resistance *Pita2* gene encoding the NB-ARC domain of blast-resistant proteins in upland rice lines, which were obtained from breeding landrace varieties. This will provide opportunities for a new source of blast-resistant germplasm for application in future upland rice breeding programs.

## 2. Materials and Methods

The experiment was conducted from March 2020 to April 2021. Molecular analysis was conducted in the Biotechnology Laboratory of Crop Production and the Biology Laboratory of Mathematics and Natural Sciences, University of Bengkulu. Screening of blast resistance was conducted in the greenhouse of the Plant Protection Department, Faculty of Agriculture, University of Bengkulu. Field evaluation was conducted in Desa Aur Gading Vill, Kerkap District, North Bengkulu Regency. The material consisted of 19 selected lines having good agronomic traits and high yield potential from the landrace breeding program, Situ Patenggang and Kencana Bali, respectively, as resistant and sensitive control varieties for checks (Table S1).

### 2.1. Screening for Blast Resistance

Leaf samples were taken in the field, which was located in North Bengkulu (BU-022), Central of Bengkulu (BT-021), City of Bengkulu (RM-023), and South Bengkulu (BS-024). Isolation was performed by taking samples of plant parts (leaves or panicles) with symptoms in the form of typical lesions. Samples were cut into 1–2 cm long sections and placed on moistened filter paper in a Petri dish containing sterile water, followed by incubation at room temperature for 24–48 h to stimulate *P. oryzae* sporulation. After 24–48 h of incubation, the gray part of the sample was slowly smeared onto the surface of 4% WA medium. Isolation was carried out for monoconidia or single conidia using a needle mounted on a microscope (Figure 1d). After 4–5 days, the fungal hyphae were transferred onto Petri dishes containing potato dextrose agar (PDA) media for further propagation. Plant inoculation was performed by evenly spraying the spore suspension onto rice seedlings aged 18–21 days (having 3–4 leaves). Plants that were inoculated were stored in a humid room for 24 h, and then placed in a screen and condensed to maintain environmental humidity. Observation of the disease scale was performed using the Standard Evaluation System IRRI [33] at 7 days after inoculation (HSI) (Table S2). Observations were made regarding the latent period, number of lesions, percentage of lesion, and disease severity level.

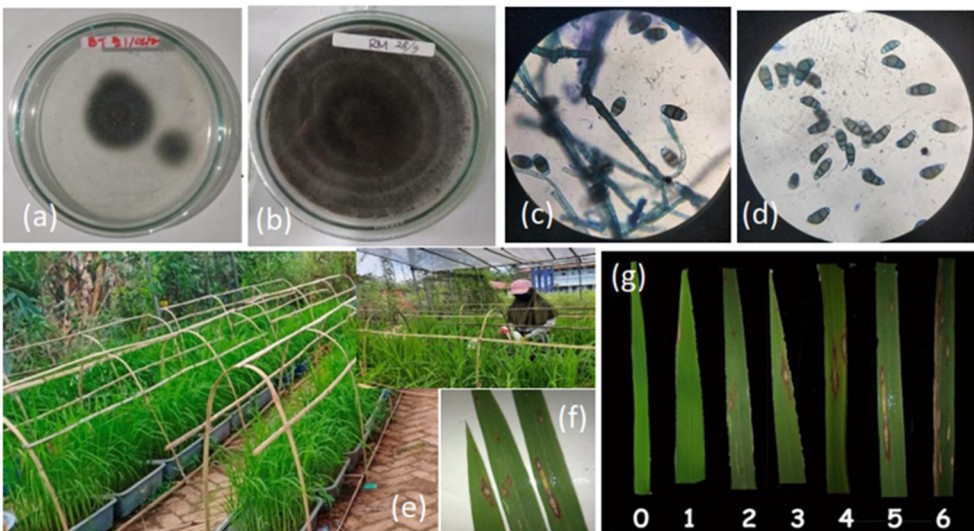

**Figure 1.** Screening for blast disease resistance of selected rice lines under greenhouse conditions: (**a**) growth of isolate in Petri dish after purification; (**b**) pure culture of blast; (**c**) microscopic identification of *P. oryzae* at 100× magnification; (**d**) isolation of monoconidia or single conidia using a needle mounted on a microscope; (**e**) greenhouse experiment; (**f**) the blast lesion lengths of susceptible line; (**g**) blast lesion scale on rice leaves, ranging from 1 to 6 base on SES IRRI modified (0: no symptom; 1: spots in the form of needle of several mm but not yet elliptical; 2–3: Elliptical shaped spots, infected leaf surface area reaches 2%; 4–5: Infected leaf surface area reaches > 2 ≤ 10%); 6–7: Infected leaf surface area reaches >10 ≤ 50%) (Table S2).

### 2.2. DNA Extraction, PCR Analysis and DNA Amplification

Fresh leaf samples were collected from plants in the greenhouse trial. The leaf samples were placed in plastic and then in a designated box filled with ice to maintain their freshness. A total of 0.1 g of rice leaf pieces was crushed by adding liquid nitrogen, and total DNA isolation was then conducted using the Promega Wizard Genomic DNA Purification Kit according to the included protocol. The leaf powder was placed in a 2 mL Eppendorf tube, and 600 µL of Nuclei Lysis Solution were added; the mixture was vortexed for 1–3 s and then heated in a 65 °C water bath for 15 min. The sample was then treated with 3 µL of RNase solution and incubated at 37 °C for 15 min. Following this, 200 µL of protein precipitation solution were added, and the mixture centrifuged for 3 min at 13,000 rpm. The supernatant was then transferred into a 1.5 mL microtube, and 600 µL of isopropanol were added followed by incubation at room temperature. Centrifugation was repeated for 1 min at room temperature. The solution was then withdrawn and allowed to dry for 15 min. After this, 100 µL of DNA rehydration solution were added, and the mixture was incubated at 65 °C for 1 h or 4 °C overnight. The total extracted DNA served as a template for gene amplification using PCR. In sensitivity checks, Situ Patenggang DNA was used as the positive control, whereas Kencana Bali DNA was used as the negative control. Six pairs of primers were used in this study—*Pib*, *Pi-37*, *Pi-d2*, *Pita2*, *Pik*, and *Pik-m*—to identify the multigenic genes in rice lines (Table S3).

The amplification procedure began with a 5-min predenaturation at 94 °C, followed by 35 cycles of denaturation at 94 °C for 1 min, annealing for 2 min, extension at 72 °C for 2 min, and final extension at 72 °C for 10 min. Electrophoresis in 1% agarose gel in TBE buffer was used to visualize the PCR products. The electrophoretic gel was immersed in 1% EtBr for 10 min, rinsed with ddH2O for 5 min, and visualized under an ultraviolet transilluminator to observe the distribution of the DNA bands. The presence of bands of a certain size in the tested lines, corresponding to specific primer sets, indicates the presence of blast-resistant genes.

### 2.3. Sequencing and Analysis Data

DNA sequencing was performed on rice lines for confirmation of *Pita2* resistance gene detection using BigDye® Terminator First Base Services in Malaysia (PT. Genetics Sains Indonesia). The sequence data were edited using BioEdit version 7.2 and aligned using ClustalX version 2.0 and Mega X version 10.2.6. Data sequences were translated into amino acids and then analyzed using the Basic Local Alignment Search Tool X (BLASTX) of NCBI (www.ncbi.nlm.nih.gov/BLAST/) (accessed on 23 January 2022) to detect the NB-ARC protein and verify homology with amino acid sequences in the GenBank database. The Conserved Domain Database (CDD) from the NCBI tool was used to identify the sequences encoding the NB-ARC protein. The phylogenetic construction was conducted using the neighbor-joining method of the MEGA X program, with 1000 bootstraps.

### 2.4. Field Evaluation of Blast Resistance

Field evaluation was conducted in Aur Gading village, an area in which blast disease is endemic. The seeds were planted in a 2 m × 3 m plot, with spacing of 20 cm × 20 cm. Fertilization was carried out three times. The first fertilization of the plants was carried out 21–25 days after planting (DAP) at the following doses: urea, 200 kg/ha; TSP, 150 kg/ha; and KCl, 90 kg/ha. The second fertilization was carried out at 60–65 DAP with a 1/3 dose of urea, 200 kg/ha, and 1/2 dose of KCl, 90 kg/ha. The third fertilization was carried out at 70–75 HST with a 1/3 dose of urea, 200 kg/ha. Blast disease symptoms were observed using a scale centered on the Standard Evaluation System for Rice [33] (Table S2). The level of line resistance was determined using the following classification system: 0–2, resistant (R); 3, moderately resistant (MR); 4–6, moderately susceptible; and 7–9, susceptible (S).

The following formula was used to determine the severity of the disease:

$$DS = \frac{\sum_{i=0}^{n}(ni.vi)}{N.Z} \times 100\% \tag{1}$$

where *DS* denotes the disease severity; *v* is the score according to symptom criteria in family *I*; *ni* is the number of families attacked in the *i*-th score; *N* denotes the total clumps observed; *Z* is the highest score.

## 3. Results

### 3.1. Screening for Blast Resistance

Overall, 19 lines inoculated with four isolates, BT-021, BU-022, RM-023, and BS-024 showed varying blast resistance. The latent period, i.e., up to when the first lesion appeared in susceptibility checks, was less than 3 days for K. Bali, whereas it appeared later during checks of SP-resistant strains (Figure 2a). This is also evident from the number and percentage of lesions (Figure 2b,c). This shows that the susceptible varieties have a high degree of susceptibility to blast disease. Several rice cultivars showed resistance to blast disease, with lower severity observed for G7, G8, G9, G11, G13, G14, 218 G15, and G18, although G11 was sensitive to isolates RM-023, BS-024. This showed that the blast race has genetic diversity that causes different responses in plants (Figure 3). Interestingly, isolates BU-022 and BS-024 tended to be more virulent than BT-021 and RM-023. It can be seen from the heat map based on the severity of the 19 upland rice lines, where high values are shown in red, and low values are shown in yellow (Figure 4).

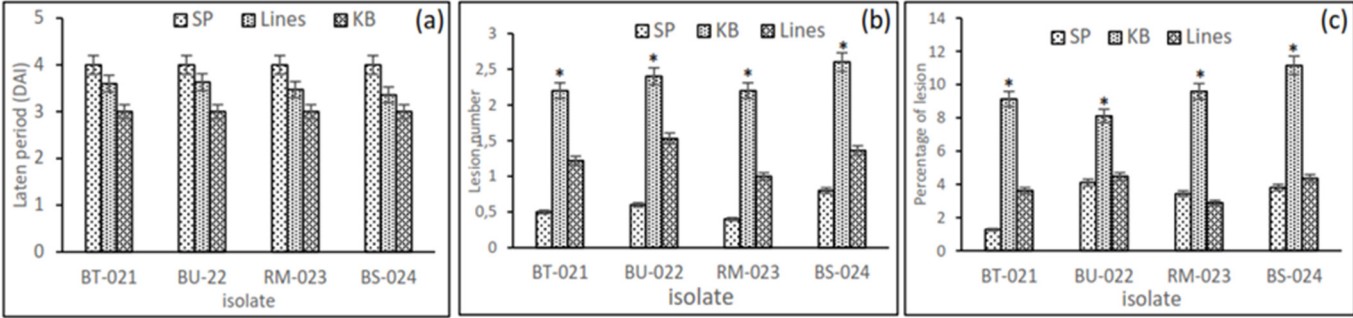

**Figure 2.** Screening for blast resistance in an inbred upland rice line using four isolates, BT-021, BU-022, RM-023, and BS-024, under greenhouse conditions: (**a**) the latent period, namely the time until lesions first appeared in the tested lines and the SP-resistant and KB-susceptible control varieties; (**b**) the number of leaf lesions on the tested lines and the SP-resistant and KB-susceptible control varieties. "*" indicate statistical significance by two-tailed Student's *t*-tests ($p < 0.05$); (**c**) percentage of lesions appearing on leaves counted in one clump

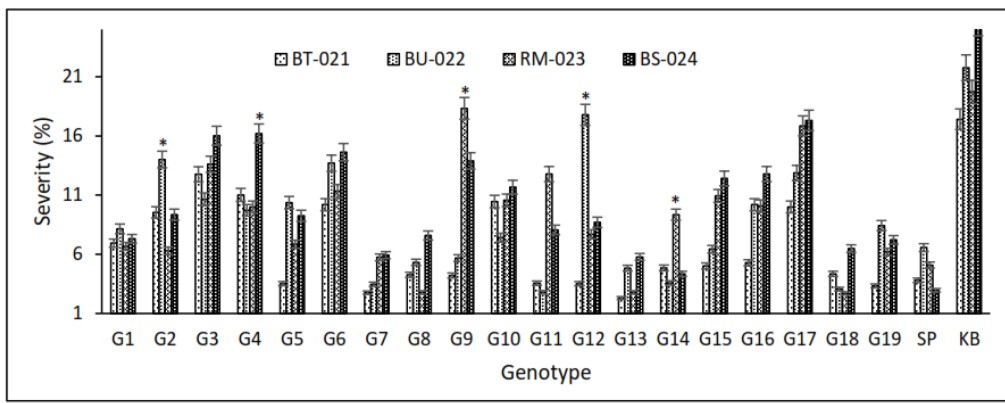

**Figure 3.** Expression of severity in 19 inbred upland rice lines, SP and KB as resistant and susceptible control varieties, respectively, which were inoculated with four isolates, BT-021 and BU-022, RM-023, and BS-024. Asterisks indicate statistical significance by two-tailed Student's *t*-tests ($p < 0.05$).

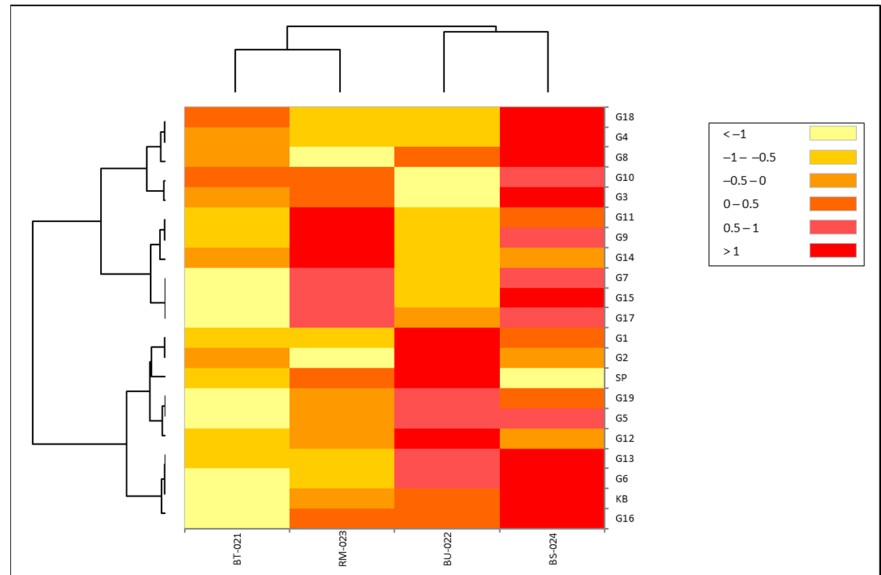

**Figure 4.** Heat map to show the virulence of isolates BT-021, BU-022, RM-023, and BS-024 based on severity level on 19 upland rice lines. High values are indicated in red and low values in yellow.

### 3.2. Detection of Blast-Resistant Genes

Detection of blast-resistant genes was performed on 19 inbred upland rice lines using six primers specific for blast-resistant genes (Table S3). The findings indicated that the six primers could detect the presence of the genes *Pi-d2*, *Pita2*, *Pi-37*, *Pik*, *Pik-m*, and *Pib* (Figure S1). The genes *Pi37*, *Pib*, *Pid2*, *Pik-m*, and *Pik* were detected in 100% of the tested lines, with sizes 1149, 388, 1058, 171, and 226 bp, respectively. Meanwhile, the *Pita2* gene was detected in 40% of lines at a size of 1042 bp (Figure S1). It would be interesting to study this further to relate the results of the PCR assay to blast disease resistance in the greenhouse and in the field because, although the line may harbor numerous resistance genes (Figure 5), it may not be able to overcome the virulence of a specific blast race encountered in the field.

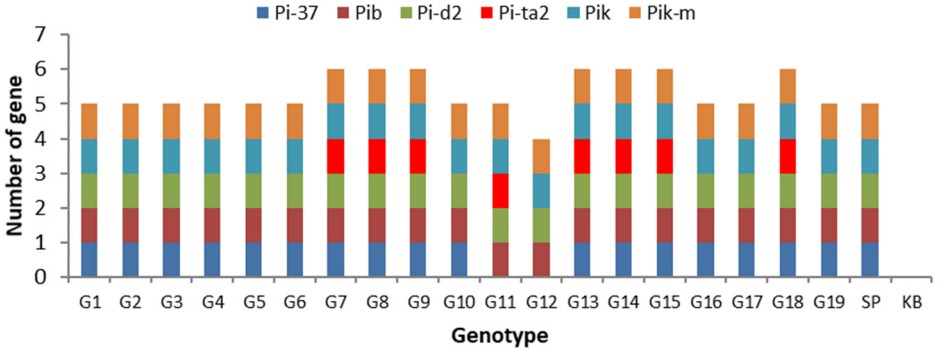

**Figure 5.** The presence of blast-resistant genes, indicated by rice lines; the red color indicates a *Pita2* gene, which is a unique protein essential for broad-spectrum disease resistance.

The PCR assay showed that eight lines, namely G7, G8, G9, G13, G14, G15, and G18, possess all examined blast-resistant genes (Figure 5). Situ Patenggang, as a resistant control, has five genes, whereas Kencana Bali, as a susceptible control, does not possess any resistance genes. The *Pita2* gene encodes a protein that is essential for broad-spectrum blast resistance, mediated by the NLR R gene. Several lines were found to carry the *Pita2* gene, namely G7, G8, G9, G11, G13, G14, G15, and G18. This is consistent with greenhouse screening, where lines expressing the *Pita2* gene tend to be more resistant to blast disease.

### 3.3. Sequence Analysis to Determine Genes Encoding the NB-ARC Domain Proteins for Blast Resistance

The Basic Local Alignment Search Tool X (BLASTX) program was used to verify the amino acid sequence homology from this study with NCBI amino acid sequences (Figure 6A). We found *Pita2* sequences in eight lines encoding the NB-ARC protein, which were not found in the other genes, although bands were positive (Figure 6B). Blast results on NCBI show in detail the putative conserved domain of the NB-ARC location with high similarity.

The conserved domain in eight blast-resistant rice lines detected the location of NA-ARC at sequence lengths between 300 and 870 (±450 bp) (Figure 6B). The results of the alignment analysis indicated that the query sequences were highly similar to 15 subjects that were homologous to the data in the gene bank (Figure 6A). The highest similarity between amino acid sequences for the *Pita2* gene detected in eight rice lines (Table S4) was found between the NBS-LRR resistance protein-partial (*Oryza sativa* indica Group), which showed 96–99% similarity, typically encoding proteins with NLR domains. Other homologies found were an NB-ARC domain containing protein-expressed (*O. sativa* japonica Group) with 96.88% similarity and blast resistance protein *Pita* variant 12 (*O. barthii*), blast resistance protein *Pita* variant 14 (*O. nivara*), and NBS-LRR-partial (*O. sativa* indica Group), with similarity of 95.3%, 93.2%, and 81.25%, respectively. Higher similarity suggests a more accurate gene sequence, where two DNA fragments may be proven to be homologous, with 70% of the base sequence or 25% of the amino acid sequence being identical (order of ≥100 bp).

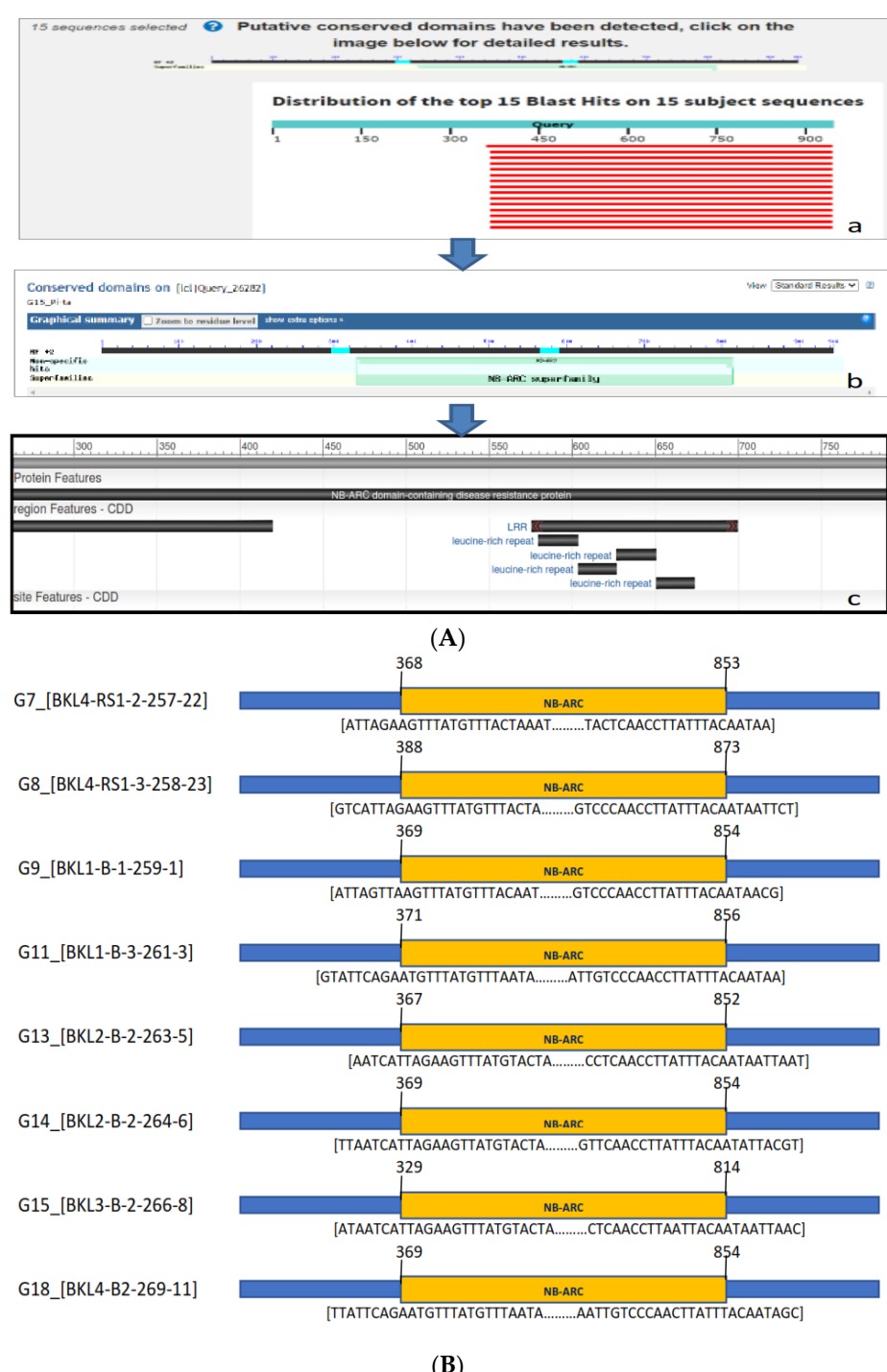

**Figure 6.** (**A**) BlastX translated nucleotide protein in the NCBI gene bank; (**a**) query sequence with 15 subjects homologous to the data in the gene bank; (**b**) the putative conserved domain detected by the query gene containing the NB-ARC superfamily; (**c**) the NBS-LRR protein has been identified as a key defense resistance mechanism in plants. (**B**) The conserved domain in eight blast-resistant rice lines detected the location of NB-ARC using BlastX. The yellow region showed the location of NB-ARC at sequence lengths between 300–870 (±450 bp).

The expected value (E-value) is a statistically computed probability value describing the sequence similarity of rice lines acquired from Gene Bank (www.ncbi.nlm.nih.gov) (accessed on 23 January 2022). Phylogenetic analysis using the neighbor-joining method revealed that the detected rice lines carrying the *Pita2* gene formed the same three groups. Figure 7

demonstrates that there were three large clusters, including cluster I, showing 97% similarity in the CCF78549.1 (NBS-LRR, partial [*O. sativa* Indica Group]) assessment, with E-value 2E-123. Cluster II has 98% similarity with the accession groups AFH58009.1, CCD33216.1, AFH54048.1, AFH54050.1, AFH54047.1, AFH54039.1, ACV87221.1, and CCD21829.1, while cluster III has similarity of 44% with the accession groups ABA97435.1, ACI49447.1, AAK00132.1, ACI49442.1, ACI49449, and ACI49451.1. BLAST investigation of the amino-acid-coding gene for the NB-ARC domain-containing protein revealed significant results.

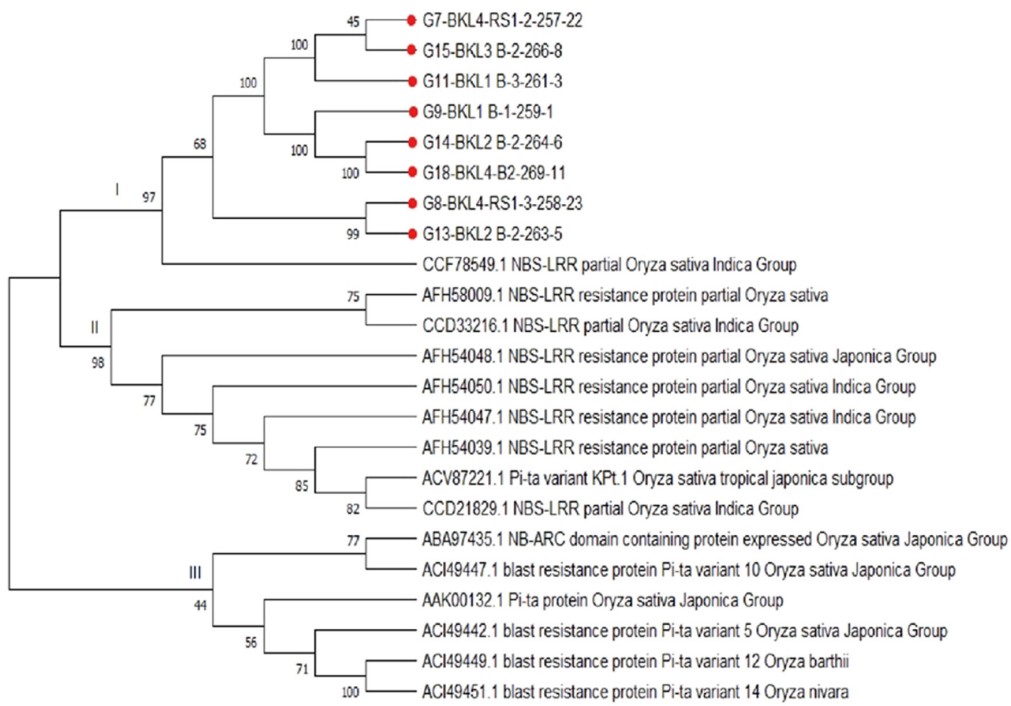

**Figure 7.** Phylogenetic tree constructed using neighbor-joining approaches and 1000 bootstrap replications to compare amino acid sequences from eight rice lines identified by the *Pita2* gene.

### 3.4. Field Evaluation of Blast Resistance

Symptoms of blast disease began to appear on the leaf margins of the plants 7 weeks after planting (WAP) (Figure 8a) and rectangular brown spots appeared at 10 WAP (Figure 8b). We did not find any explosion attacks by several line numbers until the end of the observation (Figure 8c).

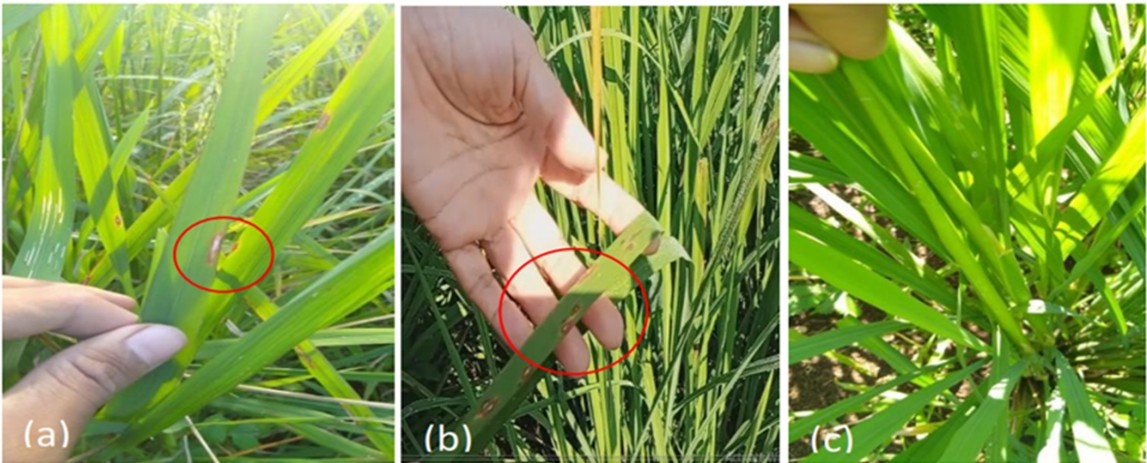

**Figure 8.** Observation of blast disease in the field; (**a**) blast symptom on the edge of the leaf; (**b**) blast attack has led to the formation of a rectangular brown spot; (**c**) resistance line

Symptoms of field blast on Kencana Bali, as a susceptible variety, with the highest scale of 6, indicate that the blast attack did not possess the severity to cause the death of the plant population. Situ Patenggang, as a resistant variety, only scored 1 (Table 1). The highest values for the G10, G16, and G17 lines, with a score of 4–5, indicated a moderate susceptibility response. Several lines showed a resistant response, with a score of 0–1, in the G7, G8, G9, G18, and G19 lines.

**Table 1.** Scores of blast disease in the field for 19 selected upland rice lines with two varieties control resistance and susceptibility, respectively.

| Genotype | Accession | The Lowest Score | The Highest Score | Mean Score | Reaction |
|---|---|---|---|---|---|
| G1 | BKL1-RS1-1-247-13 | 0 | 2 | 1.4 | R |
| G2 | BKL1-RS1-1-248-14 | 0 | 3 | 1.8 | R |
| G3 | BKL1-RS1-2-249-15 | 0 | 3 | 3.4 | MR |
| G4 | BKL2-RS1-1-251-17 | 0 | 2 | 1.8 | R |
| G5 | BKL3-RS1-1-253-18 | 2 | 3 | 3.4 | MR |
| G6 | BKL4-RS1-1-256-21 | 0 | 3 | 3.2 | MR |
| G7 | BKL4-RS1-2-257-22 | 0 | 1 | 0.4 | R |
| G8 | BKL4-RS1-3-258-23 | 0 | 2 | 1 | R |
| G9 | BKL1 B-1-259-1 | 0 | 1 | 0.8 | R |
| G10 | BKL1 B-2-260-2 | 3 | 5 | 4.6 | MS |
| G11 | BKL1 B-3-261-3 | 0 | 2 | 2.4 | R |
| G12 | BKL2 B-1-262-4 | 0 | 2 | 2.6 | R |
| G13 | BKL2 B-2-263-5 | 0 | 2 | 2.8 | R |
| G14 | BKL2 B-2-264-6 | 0 | 2 | 1.6 | R |
| G15 | BKL3 B-2-266-8 | 0 | 2 | 2.2 | R |
| G16 | BKL3 B-3-267-9 | 2 | 4 | 4.2 | MS |
| G17 | BKL4 B-1-268-10 | 4 | 4 | 5 | MS |
| G18 | BKL4-B2-269-11 | 0 | 1 | 0.6 | R |
| G19 | BKL4 B-3-270-12 | 0 | 1 | 0.6 | R |
| G20 | Kencana Bali | 5 | 6 | 5.4 | MS |
| G21 | Situ Patenggang | 0 | 1 | 0.6 | R |

The observations were performed on 10 samples of each line number; blast disease was observed based on the Standard Rice Evaluation System (IRRI, 2013). Determination of resistance used the following scale: 0–2, resistant (R); 3, moderately resistant (MR); 4–6, moderately susceptible (MS); and 7–9, susceptible (S).

All lines showed a level of disease severity below that of the susceptible variety, Kencana Bali (62%) (Figure 9A), with severity ranging from 30% to 49%, i.e., the G5, G6, G10, G16, and G17 lines had scores between 4 and 5, indicating moderate susceptibility based on IRRI SES (Table 1). Several lines had a score of 1, equal to Situ Patenggang, with severity <20%, namely G1, G4, G7, G8, G9, G11, G12, G13, G14, G15, G18, and G19 (Figure 9A). We found that there was consistency in the PCR assay for *Pita2* gene detection, which tended to be associated with high resistance in the field and in the greenhouse conditions, namely being found in the G7, G8, G9, G11, G13, G14, G15, and G18 lines (Figure 9B).

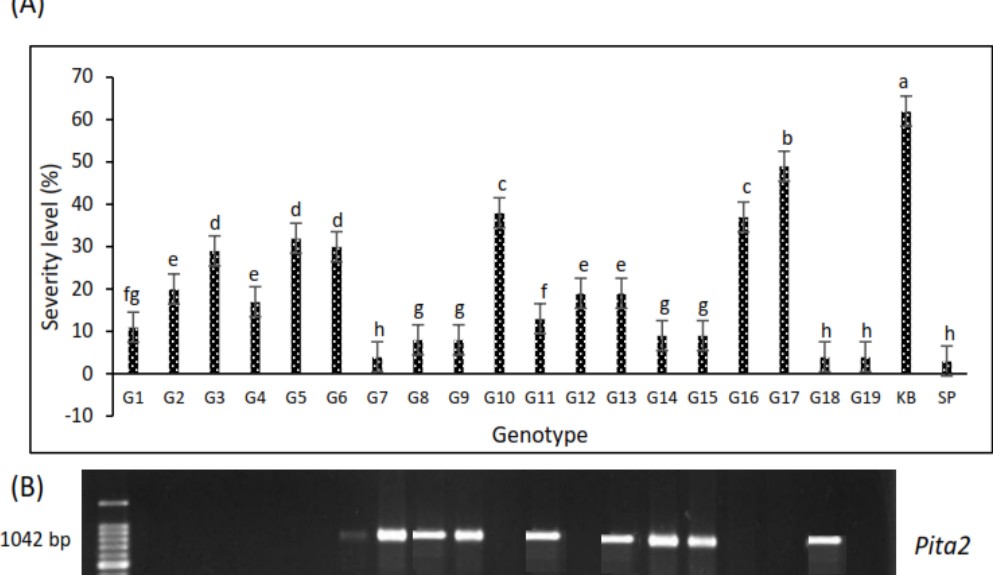

**Figure 9.** (**A**) The severity level of blast disease in the field in 19 genotypes, with Kencana Bali as a sensitive and Situ Patenggang as a resistant control. Different letters indicate statistically significant differences at $p = 0.05$ by LSD test; (**B**) the expression of the *Pita2* gene in the G7, G8, G9, G11, G13, G14, G15, and G18 lines amplified at 1042 bp showed the consistency of blast resistance in the field.

## 4. Discussion

The high genetic diversity of the race, the occurrence of race changes, and the virulence of the fungus *P. oryzae* are factors that determine the resistance of rice varieties and whether they can and how readily they can develop blast disease [6,34]. Varieties that have certain resistance genes must be adapted to the composition of the *P. oryzae* race in an area or specific location before planting [34]. Therefore, the cultivation of resistant varieties must be supported by data on the *P. oryzae* race composition of an area. Thus, it is necessary to monitor *P. oryzae* races in each rice agroecosystem, especially in areas where blast disease is endemic. The development of blast-resistant rice varieties that are durable and have horizontal resistance needs to be performed. One strategy to develop blast-resistant varieties can be implemented, among others, by forming genotypes through pyramiding genes or multiple resistance genes, so as to overcome the variety of blast races that develop in the field [3,4]. Previous researchers have successfully carried out a pyramid program to obtain more durable blast resistance [35–38].

The availability of germplasm resources representing high genetic diversity, through their collection and maintenance, is important in supporting rice breeding programs. Landrace varieties have been tested for their resistance to various environmental stresses, as well as pests and diseases, so they are a valuable pool of genetic resources. Genes resistant to various stresses of local varieties can be used to improve a variety through plant breeding. We have carried out a series of research studies through a drought-tolerant and blast-resistant upland rice breeding program using local Sriwijaya and Bugis and the IR 7858-1/IR148 line as gene donors for blast disease resistance [7–9]. Nineteen lines with good agronomic traits and high yield potential were selected for the detection of broad-spectrum blast resistance in this study. We used a combination of experiments, namely screening in a greenhouse, PCR assay, and field observation, and local isolates obtained from blast-endemic areas. Inoculation of isolates under greenhouse conditions showed the varying resistance of the lines, which was nonetheless higher, overall, than that of the susceptible control K. Bali (Figure 3). The latent period until the first lesion appears was the same as for resistant lines and Situpatenggang (Figure 2a), while the highest percentage of lesions on leaves was found in K. Bali (Figure 2c). Disease severity levels below 10% were found in the G7, G8, G9, G11, G13, G14, G15, and G18 lines. Although it was found

that G11 was sensitive to isolates RM-023, and BS-024, in the field, the severity was less than 20% (Figure 9A). It was observed in this study that isolates BU-022 and BS-24 are more virulent than isolates BT-021 and RM-023; it is noted that isolates BU-022 and BS-24 were obtained from an endemic area of blast, where a large number of farmers cultivate upland rice. It is thus necessary to conduct further studies on the virulence of local isolates in Bengkulu Province. Currently, we are conducting research on the virulence of various local isolates in differential varieties, with confirmation by PCR assay.

Field assessment showed that several lines had blast attacks that were less than 20% severity at a scale of 0–2, namely G1, G4, G7, G8, G9, G11, G12, G13, G14, G15, G18, and G19. The results of the blast-resistant gene detection revealed that the lines had 5–6 resistance genes, namely *Pi-d2*, *Pita2*, *Pi-37*, *Pik*, *Pik-m*, and *Pib*, which corresponded highly with having scores of 0–2 in the field blast observations. Generally, varieties with only a single resistance gene outgrow emerging lethal races [39,40], and the presence of additional major resistance genes in plants will confer broad-spectrum resistance for a longer duration [35–38]. The results of this study also indicate that plants have different degrees of resistance, confirming that there is a varying ability of plants to overcome blast attacks. Molecular detection via PCR assay revealed that not all tested lines possess the *Pita2* gene, which showed high concordance with the field blast observations, whereby the lines in which the *Pita2* gene was detected, namely G7, G8, G9, G11, G13, G14, G15, and G18, had a disease severity level of less than 10% and scores of 0–2, which is a very interesting finding. Gene pyramiding is widely regarded as an effective method for developing varieties with broad-spectrum and long-lasting resistance. A previous study demonstrated that resistance genes such as *Pi1*, *Pi5*, *Piz-5*, *Pita*, and *Pi-gm* could be improved into elite cultivars through marker-assisted selection [11,41].

We performed sequencing on the lines that were shown to carry the *Pita2* gene. Therefore, further studies were conducted with sequencing of the eight lines to determine the expression of genes encoding the NB-ARC domain of blast-resistant proteins. Conserved domain analysis revealed that eight blast-resistant rice lines were found to encode NB-ARC at sequence lengths between 300 to 870, an association that was considered very significant at the lowest E-value of $8.45 \times 10^{-11}$. The BLASTX program was used to determine the homology of amino acid sequences from this study with those from NCBI databases. Sequence analysis via BLASTX revealed that there were 15 gene homologs of the eight rice lines detected as the *Pita2* gene. The highest similarity between amino acid sequences for the *Pita2* gene detected in eight rice lines was observed for the NBS-LRR resistance protein-partial (*Oryza sativa* Indica group), with a similarity level of 81–99%. The result is consistent with the phylogenetic tree, which demonstrates that the groups of homologous genes are closely related to the sequences of eight lines. In addition, the *Pita* protein [*Oryza sativa* Japonica group], blast resistance protein *Pita* variant 5 [*Oryza sativa* Japonica group], and NBS-LRR partial [*Oryza sativa* Indica group] had the lowest associated E values, which were between $7 \times 10^{-11}$ and $8 \times 10^{-11}$ The BLAST analysis for the gene coding for the NB-ARC domain-containing protein showed a significant identity. The E-value for the BLAST analysis is considered to be significant if the in-between value is $1 \times 10^{-10}$ or lower [42].

The phylogeny also demonstrated that the sequences of eight lines formed three clusters. Moreover, the alignment analysis in this study showed that the query sequences had very high similarity with 15 homologs in GenBank. The putative conserved domain results identified the query gene sequence as containing a domain corresponding to the NA-ARC superfamily. Furthermore, the NBS-LRR resistance gene has been expressed in the largest group of plant resistance genes, with a key role in plant defense responses. The immune system in plants depends upon the role of encoded proteins, and it is revealed that these genes are more highly conserved than others. Plants encode resistance proteins (R) through pathogen gene-for-gene recognition by inducing defense hypersensitivity responses to protect against pathogen infection [43].

The R protein is found between APAF-1 and CED-4 on the plant-conserved ARC domain. Through nucleotide bond formation with P-loop and Walker motifs, ARC1 contains a four helical bundle and ARC2 has a winged helical fold, together constructing the three subdomains of the NB-ARC domain [44,45]. Signal initiation is triggered by the ARC domain, which translates the modulated elicitor from the C-terminal [46].

We confirmed the presence of the gene encoding the NB-ARC domain of novel blast resistance proteins in eight new rice lines from the breeding program aimed at the development of landrace varieties. The study also revealed that these lines possess polygenic genes with the potential for broad-spectrum blast resistance. Further studies on the performance and resistance of lines in local blast race isolates and field trials in various blast-endemic areas will also provide valuable information about their potential as candidates for new rice varieties. The blast-resistant genes could provide genetic resources for breeding programs that are needed to develop rice varieties with longer-lasting resistance and long-term resistance.

## 5. Conclusions

In this study, eight rice accessions were found to show complete resistance to a blast based on a field evaluation and greenhouse conditions. The molecular analysis showed that the lines had numerous genes, where the *Pita2* gene was the target of the DNA sequencing for the analysis of genes encoding the NB-ARC domain of blast resistance proteins at sequence lengths between 329 and 873. This study revealed that resistance gene introgression in the landrace germplasm could be a source of blast-resistant genes to develop new varieties. The eight lines analyzed in this study, namely G7, G8, G9, G11, G13, G15, and G18, have polygenic resistance and a potential as a novel genetic resource in the program for breeding various blast-resistant upland rice cultivars to overcome infection by blast pathogens that have multiple races and varied dynamics. Further comprehensive studies should be performed to confirm the performance and resistance of the candidate lines in field trials at various blast-endemic areas before they can be released as new rice varieties.

**Supplementary Materials:** The following supporting information can be downloaded at: https://www.mdpi.com/article/10.3390/agronomy12102373/s1, Table S1: Selected lines resulting from crossbred for identification of blast resistance; Table S2: The scale of blast disease symptoms for field assessment based on SES IRRI; Table S3: Primer characteristics for identifying blast resistance; Table S4: Analysis of gene homology using BLASTX; Figure S1: PCR assay showed blast resistance gene expression in 19 upland rice lines using specific primers Pi37, Pib, Pi-d2, Pi-ta2, Pik-m, and Pik (M = 100 bp marker, SP = Situ Patenggang, KB = Kencana Bali; 1–19 = line numbers).

**Author Contributions:** Conceptualization, R.H. and S.H.; methodology, R.H. and H.B.; software, D.W.G. and S.; validation, R.H., S.H. and H.B.; formal analysis, R.H. and S.; investigation, H.B.; resources, R.H., S.H. and H.B.; data curation, D.W.G.; writing—original draft preparation, R.H. and S.H.; writing—review and editing, D.W.G.; S. and S.H.; visualization, H.B. and D.W.G.; supervision, R.H.; project administration, R.H.; funding acquisition, R.H. All authors have read and agreed to the published version of the manuscript.

**Funding:** This research received no external funding.

**Institutional Review Board Statement:** Not applicable.

**Informed Consent Statement:** Not applicable.

**Data Availability Statement:** The data presented in this study are available on request from the corresponding author.

**Acknowledgments:** This research was funded by the PNBP Faculty of Agriculture, University of Bengkulu (contract No. 590/UN30.11/LT/2021). The authors would like to thank Aji Satrio and Ahmad Zubaedi for their assistance in field experiments, and Peni Wahyuni and Nurul Hamidah for their assistance in the greenhouse. We are grateful to the Head of Research and Community Board, Dean of the Agricultural Faculty, and Head of the Department of Crop Production at the University of Bengkulu to facilitate this research.

**Conflicts of Interest:** The authors declare that no conflict of interest exists.

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
