# Peer review of "Improving Broad Spectrum Blast Resistance by Introduction of the Pita2 Gene: Encoding the NB-ARC Domain of Blast-Resistant Proteins into Upland Rice Breeding Programs"

_agronomy, doi:10.3390/agronomy12102373_

Round 1

Reviewer 1 Report

1.     The title is too long, it is suggested that the title could be: “Improving blast resistance by introduction pita2 gene into upland breeding program.” Or “pita2 is an effective genetic resource for blast resistance in upland rice breeding program”

2.     In line 47-49, the information of 2007 is too old, please update it if possible.

3.     In lines 217-218, “Several rice cultivars showed resistance to blast disease, with lower severity observed for G7, G8, G9, G11, G13, G14, 218 G15, and G18 (Figure 3).” However, in figure 3, the severity of G9 by RM-023, BS-024 were not so good compared the average of tested 18 lines, please revise or rewrite it.

4.     Lines 315-317, “Sequencing analysis was conducted to determine the genes encoding the NB-ARC domain of blast-resistant proteins in new upland rice lines to detect broad- spectrum blast resistance.” could be delete.

5.     Lines 374, 377, 378, 379, 393, 509, 512, 513 and figure 7, “Indica” should be “indica”, “Japonica” should be “japonica”, and should be in italic type. “Oryza nivara” should be in italic type.

6.     In table 1, in the header row, what parameter was presented in the column of “mean”? please specify it.

7.     Lines 473-474, “Disease severity levels below 10% were found in the G7, G8, G9, 473 G11, G13, G14, G15, and G18 lines (Figure 3).”  Please rewrite it as the severity of G9 by RM-023, and BS-024 is higher than 10%.

8.     Consider adding a data based comparison to compare the severity of rice blast between groups with and without pita2, so as to emphasize the importance of pita gene to rice blast resistance, for example, the t-test between the severity of the two groups of rice blast.

9.     Line 475, “BT-022”, could be “BT-021”?

10.  In the "Introduction" section, please add more information about pita2. In the discussion section, please write succinctly.

11. Grammar and Spell Checking, such as: line 59, “[5, 6]” should be “[5, 6].”; line 60, “parent” should be “parents”; line 61, progenyshould be “progenies”; line 62, “establi shment” should be “establishment”; line 68 “molecular” should be “. Molecular” etc.

Author Response

Dear Reviewers,

We would like to express our appreciation to you and the anonymous reviewer for the time and effort that had been spent in review our paper. We confirm that the paper has been appropriately revised in accordance with your comment.  With pleasure, please find the file in the attachment.

Once again, sincere thanks for the time and effort in further our revised manuscript.

Sincerely,                                                                                  

Reny Herawati_et al

Reviewer 2 Report

In the submitted manuscript, Reny Herawati et al. detected eight positive lines exhibiting complete resistance to blast using field evaluation and greenhouse conditions and found Pita2, encoding the NB-ARC domain of blast resistance proteins, could be the target gene for this resistance, which is the benefit to developing new blast-resistant upland rice varieties in the future.

My comments are as follows:

1. What is 1-6 mean in Fig.1g? The author should provide detailed information.

2. Fig. 2 and 3 should provide the statistical difference analysis.

3. The author should provide the number for the high and low values in the heat map in Fig.4.

4. Did the author develop the over-expression line and mutant lines of Pita2, the phenotype of these lines would strongly support their conclusion. 

Author Response

Dear Reviewers,

We would like to express our appreciation to you and the anonymous reviewer for the time and effort that had been spent in review our paper. We confirm that the paper has been appropriately revised in accordance with your comment.  With pleasure, please find the file in the attachment.

Once again, sincere thanks for the time and effort in further our revised manuscript.

Sincerely,                                                                                  

Reny Herawati_et al

Response to Reviewer 2 Comments:

First of all, we deeply appreciate your helpful comments. Our replies to the reviewer’s inquiries and revised points are as follows. Please confirm the relevant parts highlighted in red in the revised manuscript.

Point 1: What is 1-6 mean in Fig.1g? The author should provide detailed information.

Response 1:  Thank you for your valuable comments. We have added sentences to describe Fig. 1g: blast lesion scale on rice leaves (0: no symptom; 1: spots in the form of needle of several mm but not yet elliptical; 2-3: Elliptical shaped spots, infected leaf surface area reaches 2%; 4-5: Infected leaf surface area reaches > 2 -- < 10%); 6-7: Infected leaf surface area reaches > 10--< 50%)

Point 2: Fig. 2 and 3 should provide the statistical difference analysis.

Response 2: Thank you for your valuable comment. As the reviewer suggested, we have added the statistical analysis using two-tailed Studen’s t-tests (P<0,05) for fig 2 and 3.

Point 3: The author should provide the number for the high and low values in the heat map in Fig.4.

Response 3: Thank you for your valuable comment. As the reviewer suggested, we have added the number for the high and low values in the heat map in Fig.4.

Point 4: Did the author develop the over-expression line and mutant lines of Pita2, the phenotype of these lines would strongly support their conclusion.

Response 4: Thank you for your valuable comment. We have not yet developed the over-expression line and the Pita2 mutant line, furthermore, we will focus on developing these lines in the blast endemic area, to obtain accurate yield information with good phenotyping performance.

Round 2

Reviewer 2 Report

The authors have improved the manuscript considerably, and I think the MS can be accepted.